# Treatment Outcome Prediction Using Multi-Task Learning: Application to Botulinum Toxin in Gait Rehabilitation

**DOI:** 10.3390/s22218452

**Published:** 2022-11-03

**Authors:** Adil Khan, Antoine Hazart, Omar Galarraga, Sonia Garcia-Salicetti, Vincent Vigneron

**Affiliations:** 1Informatique, Bio-Informatique et Systèmes Complexes (IBISC) EA 4526, Univ Evry, Université Paris-Saclay, 91020 Evry, France; 2Department of Computer Science, Sukkur IBA University, Sukkur 65200, Sindh, Pakistan; 3UGECAM Ile-de-France, Movement Analysis Laboratory, 77170 Coubert, France; 4SAMOVAR, Télécom SudParis, Institut Polytechnique de Paris, 91764 Palaiseau, France

**Keywords:** multi-task learning, clinical gait analysis, gait rehabilitation, deep learning, long short-term memory, botulinum toxin

## Abstract

We propose a framework for optimizing personalized treatment outcomes for patients with neurological diseases. A typical consequence of such diseases is gait disorders, partially explained by command and muscle tone problems associated with spasticity. Intramuscular injection of botulinum toxin type A is a common treatment for spasticity. According to the patient’s profile, offering the optimal treatment combined with the highest possible benefit-risk ratio is important. For the prediction of knee and ankle kinematics after botulinum toxin type A (BTX-A) treatment, we propose: (1) a regression strategy based on a multi-task architecture composed of LSTM models; (2) to introduce medical treatment data (MTD) for context modeling; and (3) a gating mechanism to model treatment interaction more efficiently. The proposed models were compared with and without metadata describing treatments and with serial models. Multi-task learning (MTL) achieved the lowest root-mean-squared error (RMSE) (5.60°) for traumatic brain injury (TBI) patients on knee trajectories and the lowest RMSE (3.77°) for cerebral palsy (CP) patients on ankle trajectories, with only a difference of 5.60° between actual and predicted. Overall, the best RMSE ranged from 5.24° to 6.24° for the MTL models. To the best of our knowledge, this is the first time that MTL has been used for post-treatment gait trajectory prediction. The MTL models outperformed the serial models, particularly when introducing treatment metadata. The gating mechanism is efficient in modeling treatment interaction and improving trajectory prediction.

## 1. Introduction

Fatigue, weakness, sensory loss, ataxia, and spasticity are among the usual causes of motor impairments due to neurological diseases such as multiple sclerosis (MS) [1], TBI, spinal cord injury (SCI), and CP, among others. For this reason, physicians often advise people with such impairments to be treated in rehabilitation as a supplement to their background pharmacologic treatment. Spasticity is a motor disorder characterized by a velocity-dependent increase in tonic stretch reflexes (muscle tone) with exaggerated tendon jerks resulting from the hyper-excitability of the stretch reflexes as one component of upper motor neuron syndrome [2]. Intramuscular injection of BTX-A is a standard treatment for spasticity. It has been shown that BTX-A produces improvements in lower and upper limb function [3], thereby improving movement such as walking [4] (see Figure 1) or fine motor skills. The minimum and maximum dose of BTX-A may vary depending on the muscle that is considered [5]. Furthermore, the total dose of BTX-A (sum of doses for all treated muscles) should not exceed the recommended amount according to the patient and the considered muscles (i.e., upper limbs and lower limbs).

BTX-A is a relatively expensive pharmaceutical product, and its consumption has increased in recent years [6,7]. Although its effect on muscle function is considered reversible, BTX-A treatment presents risks (i.e., undesirable effects), and injection sessions should be spaced by at least 3 months apart. For all these reasons, optimizing BTX-A treatment by choosing the right muscles to be treated and the dose distribution is a complex task of great relevance and requires careful study of the patient’s condition.

In practice, decision-making is based on a patient’s medical history, physical examination, and clinical movement analysis (CMA). CMA consists of studying movement troubles and identifying their plausible causes, based on bio-mechanical interpretation of instrumental measures [8] (Figure 2). If certain quality criteria are fulfilled, CMA data are sufficiently reliable for clinical interpretation [9]. CMA techniques can be used to analyze lower limb movement (e.g., walking, climbing stairs, running, etc.) or fine motor skills. Numerous scientific studies have shown that CMA, especially clinical gait analysis (CGA), provides considerable aid in the assessment and treatment decision for various neurological diseases such as CP [10], post-stroke hemiparesis [4], and MS [11], among others. Camardella et al. [12] support the idea of using Machine Learning to also predict clinical scores after robot-assisted rehabilitation as a decision-support tool for clinicians.

Artificial intelligence (AI) and machine learning (ML) techniques have become almost ubiquitous in our daily lives by guiding our decisions and providing recommendations. Therefore, it is not surprising that ML approaches are becoming increasingly popular in precision medicine and fulfill an increasing demand for new healthcare solutions, in particular a better understanding of pathological processes. Among AI and ML methods, the deep neural network (DNN) [13] has already shown spectacular results in aiding clinical decision-making [14]. The DNN requires a significant amount of data to be properly trained. However, available experimental databases are often limited in size, which makes them impractical to construct DNNs for prediction models. Medical data are often heterogeneous, complex, incomplete, uncertain, multi-modal, and multilevel, drastically decreasing the amount of exploitable data and questioning the development of prediction models [15]. Machine learning (ML) models must be able to manage data of a different nature describing the patient (images, time series, discrete clinical data, etc.) and link them to data from treatment in nominal, categorical (type of treatment) [16], and/or discrete (doses) forms. This requires that the model be taught a regression task between the data after and before BTX-A treatment. Since these treatments are often a combination of several factors (e.g., several drug injections), it is necessary to be able to model their interactions. Therefore, we propose a strategy to create multi-task DNN. Indeed, MTL can cope with sparse data problems and build a more robust model by sharing knowledge among different tasks [17]. MTL has been widely applied in ML and in the biomedical field to address the diversity of the data [17]. In the CGA literature, several works exploited deep learning (DL) for predicting gait trajectories, most of them on healthy gait. Su et al. [18] predicted gait trajectories and the five gait phases (loading response, mid-stance, terminal stance, pre-swing, and swing) with a long short-term memory (LSTM) to help in the design of exoskeletons. They employed either 10 or 30 time steps as the input for predicting the next five or ten steps. Twelve people were enrolled in their experiment, and the data were collected using attached inertial measurement units (IMUs) on their body parts. Zhu et al. [19] used an attention-based convolutional neural network (CNN)-LSTM to forecast the joint trajectories of the knee and ankle, based on lower and upper limb data, for the next 60 milliseconds. Zaroug et al. [20] constructed an LSTM auto-encoder to forecast linear acceleration and angular velocity trajectories. To predict five or ten steps into the future, they considered several lengths of input time steps (five to 40 steps) of kinematic data of six male participants. Hernandez et al. [21] proposed a hybrid network combining an LSTM with a CNN (DeepConvLSTM) to estimate kinematic trajectories, reaching an average mean absolute error (MAE) of 3.6°. Jia et al. [22] constructed a DNN for trajectory prediction using LSTM units and a feature fusion layer. This layer uses EMG and joint angle data. Liu et al. [23] built a deep spatio-temporal model composed of LSTM units to forecast two-time steps into the future, using the kinematic data of 35 subjects. More recently, Kolaghassi et al. [24] worked on the pathological gait trajectories of children with neurological disorders. They used two deep learning models, an LSTM and a CNN, to forecast hip, knee, and ankle trajectories. Note that all these studies tackled the prediction of the same gait cycle. The issue we face in this study is much more complex since it is centered on the impact of several treatments (BTX-A) on gait trajectories.

Our contribution consists of proposing a new solution to predict the BTX-A post-treatment gait trajectory of the patient, and possibly the interaction between different treatments. This solution is an MTL architecture, which alleviates the drawbacks previously mentioned: dataset size (number of patients), sample size (number of features), and feature diversity. To the best of our knowledge, this is the first time that MTL has been used for post-treatment gait trajectory prediction. This architecture comprises a collection of LSTM-shaped sub-models, arranged in parallel or series. Each sub-model is used for one treatment, and each treatment corresponds to an injected muscle. These muscles are attached to the left and right knees and ankles. This MTL model learns to map pre-treatment gait sequences to post-treatment sequences. A gating mechanism is proposed with different architectures to control the treatments’ influence on the final prediction.

Section 2 presents the data collection and their characteristics. Section 2.3 describes, more specifically, the different deep architectures used. The most prominent results are presented in Section 3. The paper ends with a conclusion and a short discussion.

## 2. Materials and Methods

### 2.1. Dataset Acquisition

Joint kinematics are typically acquired by optoelectronic systems [8] or inertial measurement unit (IMU) systems [25]. In this work, data were collected at the Movement Analysis Laboratory of Rehabilitation Center of UGECAM Coubert (France), using an optoelectronic Codamotion system consisting of four CX1 cameras at 100 Hz. All the patients in this laboratory were adults with different types of gait issues. This database consists of patients with central neural system disorders, e.g., CP, SCI, or TBI, and all patients had undergone spasticity treatment with BTX-A injections.

In this retrospective study, all the considered data were obtained from patients who participated in clinical activities. The database is composed of $N_{pat}=38$ patients that underwent CGA before and after spasticity treatment with botulinum toxin. The usage of these data was approved by the institution’s research ethics committee. The patients were informed about the research and did not oppose the utilization of their data. $N_{uni}=15$ patients (39.47%) were unilaterally affected (the right lower limb was affected in 6 of them and the left lower limb for the other 9), and $N_{bil}=23$ patients (60.53%) were bilaterally affected, which means that, in total, $N_{limbs}=61$ lower limbs had been modified. The data contain the CGA of patients before treatment, medical treatment details, and the CGA after treatment. The average age of the patients at the time of pre-treatment CGA, the time of injection, and the time of post-treatment CGA was 46.67 years old (yo), 46.76 yo, and 46.93 yo respectively. The range of age in the dataset is from 21 to 75 yo. There was approximately a 3-month gap between pre-treatment CGA and post-treatment CGA. The details of the patients are listed in Table 1. In this work, we considered injections into four muscles: soleus, gastrocnemius (medialis and lateralis), semitendinosus, and rectus femoris. We also defined a fifth category called “other muscles”, which groups all the other muscles that were treated (see Table 2). There were 28 different combinations of BTX-A injections of these four muscles. A treatment binary code vector:$$s^{j}={\left(s_{1}^{j},\cdots ,s_{c}^{j}\right)}^{T},s_{i}^{j}\in \left\{0,1\right\},i=1\cdots c$$
(*c* = 5 as shown in Table 1) was attributed to each lower limb *i*, with $s_{i}^{j}=1$ if muscle *i* was injected in limb *j*, 0 otherwise, and $d^{j}={\left(d_{1}^{j},\cdots ,d_{5}^{j}\right)}^{T},d_{i}^{j}\in \left\{0,1\right\}$ is a binary vector for the disease of patient’s limb *j*. There are five diseases: CP, MS, TBI, SCI, and stroke. ${}^{T}$ is the transpose operator.

### 2.2. Data Preparation

Kinematic data were automatically segmented into gait cycles from initial contact (IC) to terminal swing (TS), utilizing the high-pass algorithm (HPA) [26]. Then, gait cycles were re-sampled and normalized to 51 points (2% of the gait cycle) as proposed by CGA [27], so that the DL models were trained with fixed-length sequences, as illustrated in Figure 3. Mean gait cycles were computed for each limb.Combining both the pre- and post-treatment cycles of each patient led to a total of n=1622 gait strides. For any patient’s limb *j*, the input vector is an angular time series $x^{j}={\left(x_{1}^{j},\cdots ,x_{m}^{j}\right)}^{T}\in {\left[-180,+180\right]}^{m}$, and the target vector is $t^{j}={\left(t_{1}^{j},\cdots ,t_{m}^{j}\right)}^{T}$, with m=51×2=102. Let $D={\left\{x^{j},t^{j},d^{j},s^{j}\right\}}_{j=1}^{n}$ be the input–target training set.

The patient’s data consist of multiple gait cycles at the time of pre-treatment CGA and post-treatment CGA. Different trials were recorded for each patient. In one trial, there were multiple cycles of pre-treatment CGA. We extracted all the cycles of all patients and stored them. We separated a person’s right and left cycles since we considered them as different samples in the data. We performed the same procedure for post-treatment CGA data. Each pre-treatment cycle was associated with a target post-treatment cycle.

Note that the number of cycles per patient varies from one patient to another.

There is a total of 5 joints (pelvis, hip, knee, ankle, and foot) and three signals per joint in our dataset, leading to 15 signals. These three signals represent the projections of the trajectory of each joint, respectively, on the sagittal, frontal, and transverse planes. In this study, we only considered the knee and ankle on the sagittal plane, because most treatments were performed around these joints. Figure 3a,d show the sagittal plane signal (flexion/extension) of the ankle and knee for a patient’s complete trial containing multiple cycles.

Figure 3b,e show a cycle extracted from the full knee and ankle trials, respectively. Figure 3c,f show the normalized cycle in 51 points. In the end, our dataset contains 1622 samples and 210 features; the first feature represents the ID (patient name); the second to 103rd are the features of the pre-treatment CGA; then, c=5 features describe the presence or absence of botulinum toxin injection according to the muscle categories; finally, the last 102 features concern the post-treatment CGA of a patient.

An input matrix *X* and a target output matrix *Y* were constructed using the parameters of *n* training samples, *f* features (the sagittal plane of the ankle and knee), $l_{in}$ input size, and $l_{out}$ output size. Pre- and post-treatment data were centered and reduced by the standard deviation. The goal was to construct a model with g() that maps $\hat{Y}=g\left(X\right)$, where $\hat{Y}$ is a value that is very close to the actual value of *Y*.

### 2.3. Description of the Models

#### 2.3.1. Long Short-Term Memory

When training, early recurrent networks had difficulty remembering information for longer periods, such as several thousand time steps. Hochreiter et al. [28] introduced a particular memory cell capable of retaining information for long periods of time. The LSTM can read and write to its memory. More importantly, this memory never goes through an activation function. This effectively combats the [29] trailing gradient problem and makes the formation of this pattern very stable.

The original LSTM works with a series of input signals $x_{t}$. It has a so-called hidden state $h_{t}$ and cell state $c_{t}$ of the same size as $x_{t}$. The cell state $c_{t}$ is the model’s memory. The hidden state $h_{t}$ is the model’s prediction of $x_{t}$.

The LSTM equations are defined by the following set of matrix equations: (1)$$\begin{array}{ccc} A & = & h_{t}{∥}^{1}x_{t} \end{array}$$(2)$$\begin{array}{ccc} f_{t} & = & \sigma (W_{f}A+b_{f}) \end{array}$$(3)$$\begin{array}{ccc} i_{t} & = & \sigma (W_{i}A+b_{i}) \end{array}$$(4)$$\begin{array}{ccc} o_{t} & = & \sigma (W_{O}A+b_{O}) \end{array}$$(5)$$\begin{array}{ccc} d_{t} & = & \tanh(W_{d}A+b_{d}) \end{array}$$(6)$$\begin{array}{ccc} c_{t+1} & = & f_{t}\circ c_{t}+i_{t}\circ d_{t} \end{array}$$(7)$$\begin{array}{ccc} h_{t+1} & = & o_{t}\circ \tanh\left(c_{t+1}\right) \end{array}$$
where ${∥}^{1}$ is the concatenation operator, ∘ is the Hadamard product, σ is the logistic function, *W* are weight matrices, and b are biases. The basic idea is that the model takes the input $x_{t}$ and the previous prediction of the current input $h_{t}$, updates its internal memory $c_{t}$ to $c_{t+1}$, and then makes, a new prediction $h_{t+1}$ based on $c_{t+1}$, $h_{t}$, and $x_{t}$.

The original LSTM could have multiple parallel memory cells $c_{t}$, but in practice, mostly, only one memory cell is used; the description of the LSTM was limited to one $c_{t}$. All the gate functions (Equations (2)–(4)) are fully connected layers, y=f(Wx+b) with a sigmoid activation function. The data flow in the LSTM is illustrated in Figure 4).

Furthermore, the role of $h_{t}$ is not strictly fixed to be a prediction of $x_{t}$. It can be any series of predictions that is connected to the input series $x_{t}$. For example, if $x_{t}$ was the number of people who entered (or left) a building in the last hour, then $h_{t}$ could be the current number of people inside the building (with appropriate scaling, so it fits the output range [−1, 1]).

For this study, we used several variants of LSTM. The five categories of treatments are reported in Table 2: BTX-A injection of the first four muscles and the fifth category of injections in all other muscles. Each treatment is represented by an LSTM layer. Hidden states represent, according to the DL architecture used, the presence or absence of treatments by BTX-A in the five muscles.

While the LSTM is well suited to prediction tasks on time series, sometimes, knowledge about future events is necessary for the correct prediction. Therefore, the term “future” is relative to *t* and means the following data points. Of course, the next/future data points must already be known to be included in the prediction. Reference [30] identified two strategies to integrate the knowledge of future events into an LSTM model: bi-directional recurrent neural network (RNN) [31] and delayed input, the second approach consisting of delaying the signal by a delay τ:**Model 1** LSTM was used with pre-treatment CGA data and post-treatment CGA data. Treatments were not considered in this experiment. The model was implemented using five layers of LSTM units, with 51 units per layer, one unit for each point of a cycle. Note that each unit receives a pair of inputs for the knee and ankle, respectively. The final layer is fed into a dense layer of 102 neurons (2 × 51 values), which is then reshaped to obtain the desired output, as shown in Figure 5a. In this model, we initialized the values of the cell state and hidden state to 0.**Model 2** A total of five treatments, together with pre- and post-treatment CGA data, were included in this model, displayed in Figure 5b. In this architecture, the values were initialized according to the medical treatment. If one patient had muscle 1 and muscle 3 injected (Table 2), then each layer’s components of the hidden states vector in LSTM layer 1 and LSTM layer 3 are initialized to 1, and the other layers’ hidden states vector is initialized as 0. In this model, we also initialized the cell state as 0.

#### 2.3.2. Bi-Directional LSTM

The entire signal must be known for this approach. Two LSTM models were trained in parallel, one on the input series (forward) and the other on the reverse input series (backward), starting with the last input and then the next-to-last, and so on. Thus, for each *t*, there were two hidden states: $h_{1,t}$ and $h_{2,t}$ among the two available models. $h_{1,t}$ only contains information about the past, and $h_{2,t}$ only contains information about the future. Together, they have information about the whole signal, and the final prediction $f(h_{1,t},h_{2,t})$ was made using the two hidden states. This method has the disadvantage that two models must be trained; therefore, the number of parameters and the training time are doubled.

We studied the bi-directional LSTM (Bi-LSTM) architecture and considered two experiments, namely with and without MTD, as previously presented for LSTM:**Model 3** A Bi-LSTM, as depicted in Figure 6. As shown in Figure 6, the model has mainly the same structure as the previous Model 1 (same number of layers and units in each layer). The final layer’s hidden state is fed into a fully connected layer. As in Model 1, we initialized the values of the cell state and hidden state of each layer to 0.**Model 4** This model takes into account MTD in a multi-task architecture of Bi-LSTM models. Indeed, five Bi-LSTM models work in parallel while incorporating MTD as in Model 2. Each Bi-LSTM has 51 units, each receiving as input a pair for the knee and ankle, respectively. Input *X* is fed to the five Bi-LSTM sub-models, and the cell state of all such sub-models was initialized to 0. Furthermore, the hidden states of all sub-models were initialized according to the presence or absence of MTD (as discussed in Model 2). This architecture has two fully connected layers: the first layer concatenates the outputs of all the sub-models; the second maps the output of the first layer to 102 neurons as per the desired output, as shown in Figure 7a.**Model 5** This model is also a multi-task architecture of Bi-LSTM sub-models, as in Model 4. However, in this case, MTD is considered differently, with a gating mechanism. Instead of passing MTD as a hidden state of each Bi-LSTM sub-model, we incorporated them at the end of such sub-models by multiplying each sub-model’s output by its corresponding binary value of MTD. In other words, if there is any treatment, it will be used further in the model; otherwise, it will be discarded (multiplying with 0), as illustrated in Figure 7b. By performing this experiment, we wanted to assess the impact of this gating mechanism compared to MTD internal processing by each sub-model, as done in Model 4.**Models 6 and 7** 
In both models, we replaced the first fully connected layer (FC Layer 01) (see Figure 7a,b) with a convolutional layer (see Figure 7c,d), with kernel size (5,2) and stride (3,2). As there are five Bi-LSTM sub-models and each has an output of size 2 × 102, we concatenated such outputs and reshaped them into a matrix of size (10 × 102), then given as the input to the convolutional layer. Finally, the convolutional layer’s output is fed into a fully connected layer of size 102. Model 6 incorporates MTD as in Model 4, through the internal states of the sub-models. Model 7 uses the gating mechanism as in Model 5.

#### 2.3.3. Experimental Setup

CGA data consist of 1622 combination pre-treatment and post-treatment gait cycles of 38 patients. Leave-one-out cross validation was used to assess the models’ performance. For each iteration, we used 37 patients for training the model and one for testing. In the end, we took the RMSE of all tested patients for each model. The mini-batches were used throughout the training process, and the size of each batch was 16. We chose the RMSE as the loss function for optimizing the deep learning models and used the ADAM optimizer for learning. We tried different learning rates and selected the best-possible values. We report in Table 3 all details concerning the models’ hyper-parameters.

We calculated the RMSE to see how closely the predicted trajectories of the knee and ankle, $\hat{Y}$, matched the actual trajectories of the knee and ankle, Y. The following equation for the RMSE can be derived if we assume that *n* represents the number of testing samples, *f* represents the number of features, and $l_{out}$ represents the output size.
(8)$$\mathrm{RMSE}=\sqrt{\frac{1}{nfl_{out}}\sum_{i=1}^{n}\sum_{j=1}^{f}\sum_{k=1}^{l_{out}}{\left(y_{i,j,k}-{\hat{y}}_{i,j,k}\right)}^{2}}$$

We also calculated the standard error (SE) to measure the variation of the RMSE with respect to each disease and the $R^{2}$ score to check how well the data fit the regression model. SE is calculated using
(9)$$\mathrm{SE}=\frac{s}{\sqrt{n}}.$$
where *s* is the standard deviation of prediction with respect to a particular disease and *n* is the total number of patients having a particular disease. The coefficient of determination ($R^{2}$) as follows
(10)$$R^{2}=1-\frac{\sum_{i}{\left(y_{i}-{\hat{y}}_{i}\right)}^{2}}{\sum_{i}{\left(y_{i}-{\bar{y}}_{i}\right)}^{2}},$$
can be interpreted as the proportion of the variance in the dependent variable that is predictable from the independent variables (worst value −∞, best value =+1), opposite the MSE, which magnifies the error if the model outputs a very bad prediction (worst value +∞, best value =0).

We compared and evaluated the performance of the models with the use of these measures.

## 3. Results

We evaluated Models 1 to 7 on our dataset with the above-mentioned metrics and display the results in Table 4. The lowest average RMSE values and the highest $R^{2}$ scores are displayed in bold; they correspond to the best prediction model according to the diseases reported in Table 4.

From Table 4, we noticed that Model 4 outperformed other models in the prediction of post-treatment gait trajectories for patients having MS and TBI. Furthermore, Model 6 performed better for SCI patients than all other architectures. Model 7 outperformed other models of patients having stroke and CP. We noticed that, in all cases, the MTL architectures achieved better performance globally, on both knee and ankle signals.

The following two tables (Table 5 and Table 6) report the performance scores of the prediction of gait trajectories for knee and ankle, respectively. In Table 5, the best prediction for the knee angle was obtained for TBI patients by Model 5 with an average RMSE of 5.60° and $R^{2}=0.72$. Furthermore, for all diseases, the MTL architectures outperformed the others. Model 6 gave the best prediction for MS and SCI in terms of RMSE and Model 4 in terms of the $R^{2}$ score for the same pathologies. On the other hand, for stroke patients, Model 7 had the best average RMSE, and Model 6 had the best coefficient of determination. In Table 6, we notice that the best RMSE for the ankle was 3.77°, which is lower than that obtained for the knee (5.60°). However, even though the RMSE was usually lower (thus better) for the ankle, the $R^{2}$ scores were usually lower (thus worse) as well. In particular, for stroke, all the $R^{2}$ of the ankle angle were negative.

From a different perspective, the following graphs (in Figure 8 and Figure 9) illustrate the trajectories (pre-treatment, real post-treatment, predicted post-treatment of the patient, and standard course of an adult) of two patients. The Y-axis represents the ankle dorsiflexion or knee flexion, and the X-axis represents the gait cycle of a patient.

Figure 8 compares the prediction of different models on the knee and ankle joints in a patient diagnosed with CP. These figures differentiate the prediction between the MTL models and others. Figure 8a–c illustrate the predictions on the knee angles made by Model 1, Model 2, and Model 3, which are not MTL models. Figure 8d shows the corresponding prediction of Model 7, which is an MTL model. The predictions of post-treatment gait from Model 7 were better than others. In other words, it was closer than that patient’s expected post-treatment gait trajectory (average of all his/her target gait cycles in the training set). On the other hand, Figure 8e–h compare the prediction of the ankle joint of the same patient. Figure 8g,h illustrate the prediction of Model 1 and Model 3, respectively. Figure 8g,h show the predictions of Model 4 and Model 7, respectively, which are MTL models. We noticed that the predicted post-treatment trajectory in Figure 8g was better than the first two models, which were serial, and we see in Figure 8h the significant improvement of the prediction at the end of the gait cycle, between 80% and 100%, compared to Figure 8g. On this patient, the MTL models also performed better on the ankle joint.

Figure 9 compares the trajectories of the knee and ankle joints of another patient diagnosed with MS. Figure 9a,b, represent the predictions of the knee angles made by Model 1 and Model 2, which are not MTL models. Figure 9c,d represent the prediction of the knee angles made by Model 4 and Model 6, respectively, which are MTL models. We can see that MTL models had better predictions than the first two. The predicted post-treatment trajectories were closer to the real post-treatment trajectories. Last four Figure 9e–h compare the trajectories of the ankle joint. Figure 9e–g represent the prediction of Model 1, Model 2, and Model 3. Although Model 3 is not an MTL model, its predictions were much better than the first two serial models. However, the prediction of the MTL model (Model 5) in Figure 9h was better than all other models for this particular patient. In general, as proven by Table 4, Table 5 and Table 6, for almost every patient, MTL performed better.

## 4. Discussion and Conclusions

In this study, we used MTL to design an LSTM model and its variants to predict the post-treatment trajectory of adults with an abnormal gait. To the best of our knowledge, this specific prediction task, which exhibits greater inter- and intra-subject variability compared to the courses of normal adults, has not been addressed before in the literature using MTL.

To forecast the trajectories of the knee and the ankle in the sagittal plane, we used LSTM-based models. LSTM was chosen because it has been successfully applied to sequential data, and it can capture long-term dependencies through its learning [32]. To better evaluate the performance of MTL on a given problem, we also implemented serial models using LSTM. The RMSE was used to compare the results of both sorts of models. The RMSE of the MTL models was lower for all types of patients (different pathologies). The MTL models also gave the highest $R^{2}$, better explaining the total variance of the target than the serial models. The MTL models performed better than the serial models in our problem of multiple treatment combinations. MTL architectures allow introducing the medical treatment metadata into the model. Instead of performing a simple post–pre regression task, our results imply that introducing the treatment information (i.e., muscles treated by BTX-A) improves performance.

Overall, the best prediction was obtained for TBI using the Bi-LSTM with MTL (Models 4) architecture. The results in Table 4 show that there was only a 5.24° average difference in actual and predicted trajectories and $R^{2}=0.73$. The best maximum average RMSE error between actual and predicted trajectories was 6.24° for stroke patients, using the MTL architecture with gated Bi-LSTM and a convolutional layer (Model 7). For the knee and ankle, the best results were 6.75° ($R^{2}=0.80$) and 3.77° ($R^{2}=0.5$), respectively, for CP patients. The RMSE was usually higher for the knee than for the ankle despite having higher coefficients of determination. This suggests that the models were able to explain the variance of the knee angle better, but the amplitudes in the knee were higher than in the ankle. Moreover, no proposed model was able to adequately explain the variance of the ankle angle for patients with a stroke (only negative $R^{2}$ scores).

### 4.1. Comparison to Previous Works

Since this is the first time that the whole kinematic signals for knee and ankle on the sagittal plane were predicted for botulinum toxin treatment, it is difficult to compare our performance to other works. Nevertheless, we can compare our methods for the predictions of peak knee and ankle on sagittal planes reported by [4] for rectus femoris botulinum toxin injection of patients with stroke (Table 7). In this case, the $R^{2}$ score of the proposed method for stroke was better for peak knee flexion, but worse for peak ankle dorsiflexion. Since the compared models were not trained and tested with the same databases, this comparison must be taken with caution.

We also compared our performances to the predictions of the whole postoperative kinematic curves for patients with CP. Even though the proposed methods were not tested on the same databases, these performances were better than the postoperative predictions for CP reported by Galarraga et al. [16], Niiler et al. [33], and Niiler [34], as shown in Table 7.

### 4.2. Limitations

Besides the lack of external validation, the main limitation of the proposed models is the relatively small size of the database. DL models usually need large amounts of data to be properly trained. Unfortunately, this is rarely the case in biomedical databases. Another limitation of the model is that it does not consider other aspects of the patient, such as psychological factors, age, stress, and social environment, among others, which play a major role in the rehabilitation and, thus, in the treatment outcome.

### 4.3. Conclusions

It was concluded from the results that the number of patients and type of disease did not directly affect the model’s performance. More precisely, we can say that inter- and intra-subject variability affected the model’s performance more than the number of patients (samples) and type of disease. Table 1 gives a detailed description of the number of patients with each disease, and Table 4, Table 5 and Table 6 report the number of training samples. The minimum number of patients was 3 with CP and TBI diseases, while the maximum number of patients was 12 with MS disease. We noticed that the RMSE of CP patients and TBI patients were 6.00° and 5.24°, respectively. On the other hand, the RMSE of MS patients was 5.8°. This showed that having four-times more patients for a given disease than others did not significantly affect the RMSE value.

Finally, Bi-LSTM combined with MTL was highly effective at increasing the total quantity of information accessible to the model, enhancing the context provided to the algorithm. Future work will focus on MTL models with Bi-LSTM networks to exploit more precise information about treatments, such as the dose information, to further enhance the context given to the model.

## Figures and Tables

**Figure 1 sensors-22-08452-f001:**
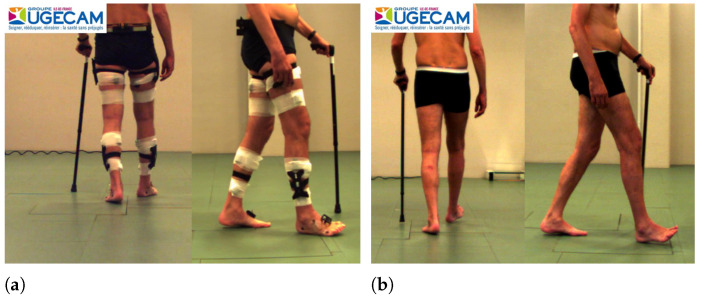
Example of the outcome of BTX-A treatment on gait (**a**) before treatment (**b**) after BTX-A treatment.

**Figure 2 sensors-22-08452-f002:**
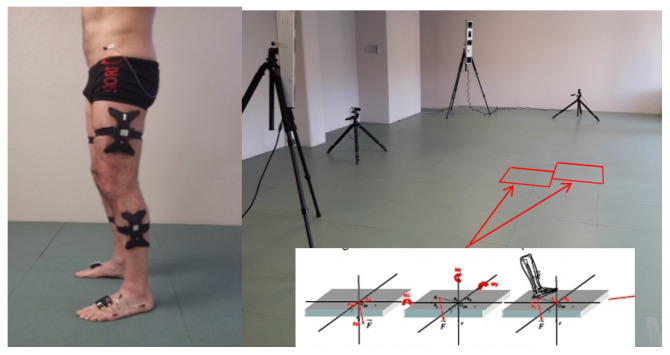
Clinical gait analysis. Different types of sensors are used to conduct kinematic and kinetic analyses of locomotion in gait labs. These may include optoelectronic motion capture, force platforms, electromyography, and IMU sensors, among others.

**Figure 3 sensors-22-08452-f003:**
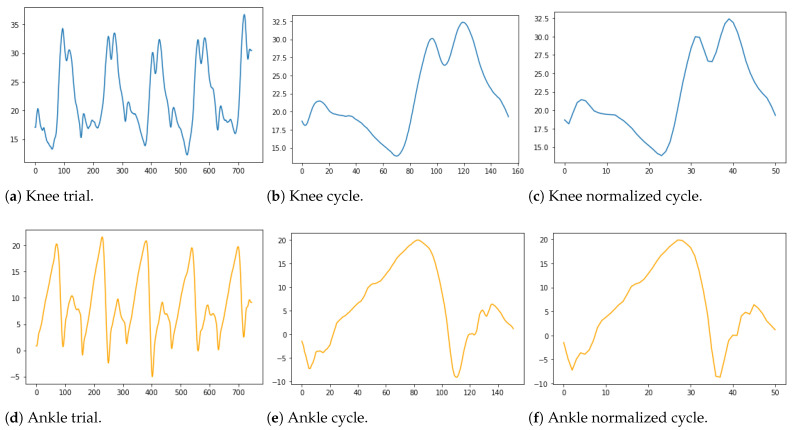
Process of converting one trial to one normalized cycle.

**Figure 4 sensors-22-08452-f004:**
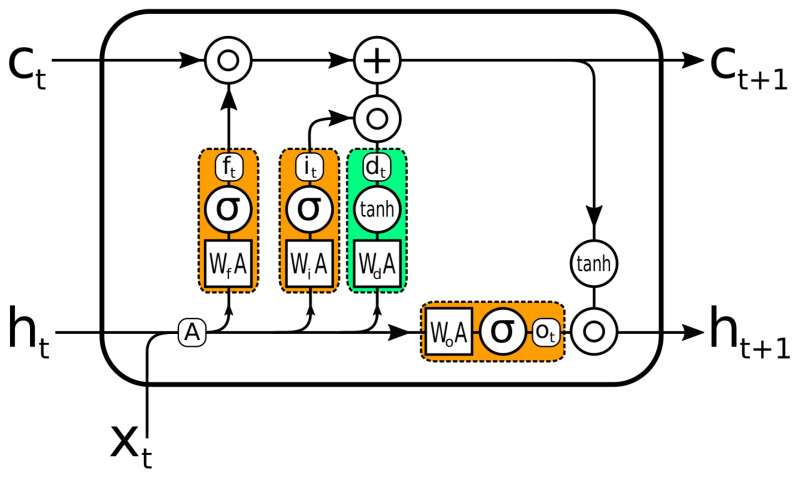
LSTM unit. The gates, which decide which part of the information to pass on, are orange. Green is the update to the memory cell.

**Figure 5 sensors-22-08452-f005:**
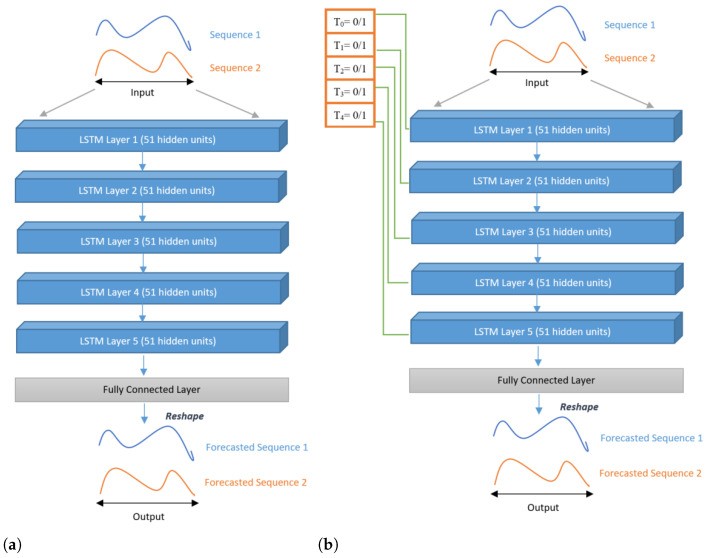
LSTM architectures (Model 1 and Model 2) proposed in this work: (**a**) without MTD; (**b**) with MTD.

**Figure 6 sensors-22-08452-f006:**
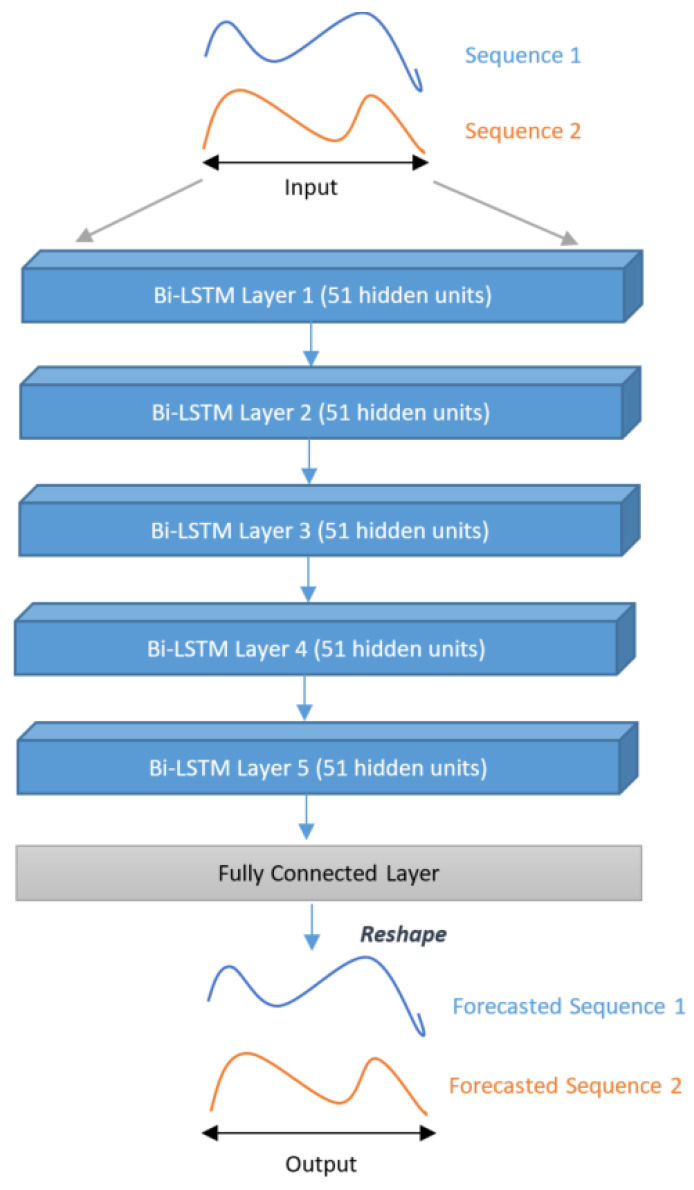
First Bi-LSTM architecture (Model 3) proposed in this work without considering MTD.

**Figure 7 sensors-22-08452-f007:**
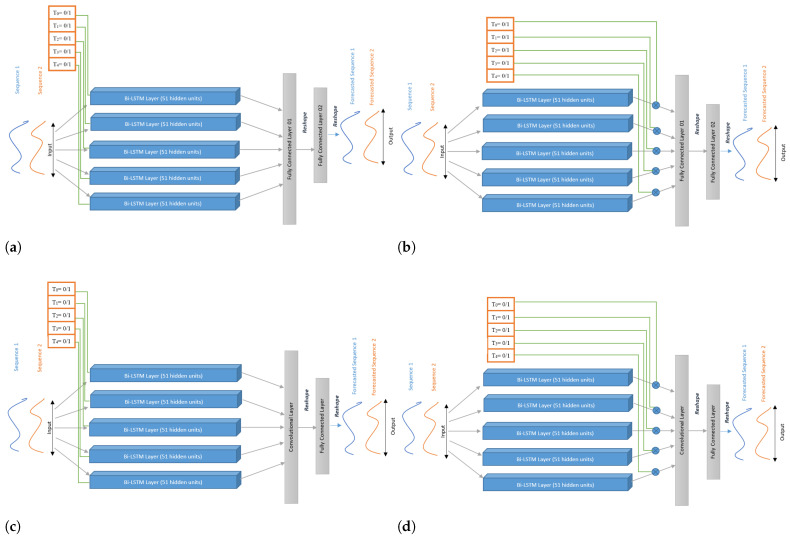
Multi-task learning architectures with Bi-LSTM sub-models; (**a**) Model 4: processing MTD internally in each sub-model; (**b**) Model 5: incorporating MTD through a gating mechanism; (**c**) Model 6: processing MTD internally in each sub-model using the Conv layer; (**d**) Model 7: incorporating MTD through a gating mechanism using the Conv layer.

**Figure 8 sensors-22-08452-f008:**
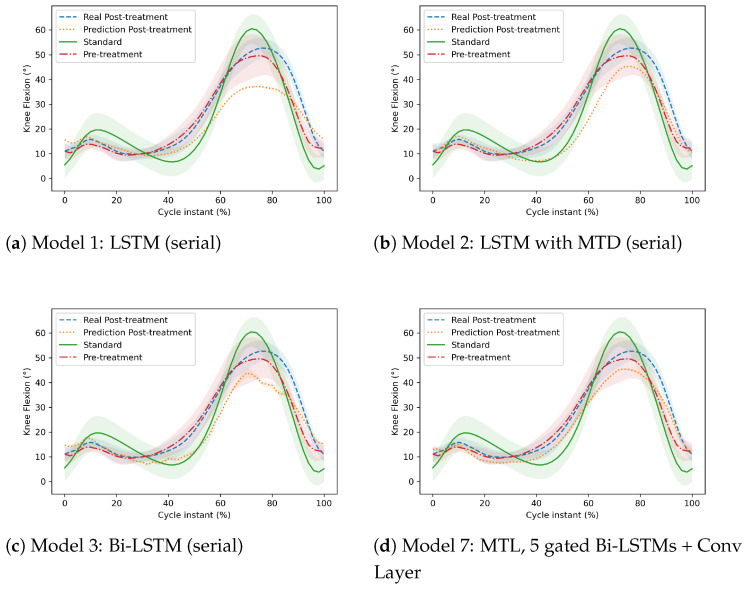
Comparison of the post-treatment gait trajectory of the knee and ankle joints in a patient diagnosed with CP. The first three models (**a**–**c**) are serial (Model 1, Model 2, and Model 3), and the fourth model (**d**) (Model 7) is an MTL model, which represents the prediction of the knee joint. The sixth and seventh models (**e**,**f**) are serial (Model 1 and Model 3), and the last two models (**g**,**h**) (Model 4 and Model 7) are MTL models, which represent the prediction of the ankle joint.

**Figure 9 sensors-22-08452-f009:**
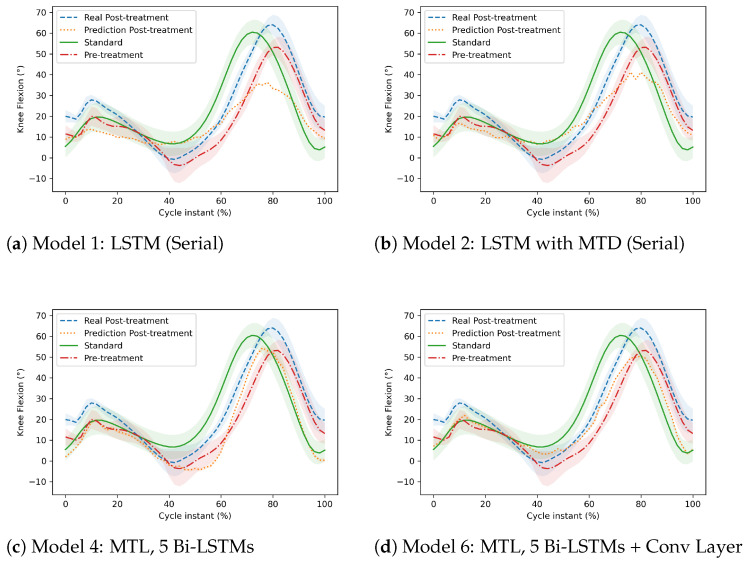
Comparison of the post-treatment gait trajectory of the knee and ankle joint in a patient diagnosed with MS. The first two models (**a**,**b**) are serial (Model 1 and Model 2), and the third and fourth models (**c**,**d**) are MTL models (Model 4 and Model 6), which represents the prediction of the knee joint. The fifth, sixth and seventh models (**e**–**g**) are serial (Model 1, Model 2, and Model 3), and the last model (**h**) is an MTL model (Model 4), which represents the prediction of the ankle joint.

**Table 1 sensors-22-08452-t001:** Patient database description.

**Total Patients**	38
**Age (Mean ± SD)**	46.76 ± 13.43
**Males/Females**	24/14
**Unilaterally/Bilaterally affected**	15/23
**Cerebral Palsy**	3
**Stroke**	9
**Multiple Sclerosis**	12
**Traumatic Brain Injury**	3
**Spinal Cord Injury**	11

**Table 2 sensors-22-08452-t002:** Considered injected muscles and their frequencies in the database.

Muscle Number	Muscle/Category	Injections in Patient	
		Number	Proportion
1	Soleus	49	29.7%
2	Gastrocnemius (Medialis and/or Lateralis)	37	28.5 %
3	Rectus Femoris	18	10.8%
4	Semitendinosus	12	7.2%
5	Other Muscle	40	24.2 %

**Table 3 sensors-22-08452-t003:** Hyper-parameter selection for LSTM, Bi-LSTM, and other variants of the architecture. The MTD column is used for medical treatment data (included/not included).

Model No. and Fig. Reference	Model Type	MTD	LSTM Layers (Units)	Conv. Layer	FC Layers (Units)	Learning Rate
Model 1 (Figure 5a)	LSTM (serial)	No	5 layers (51)	None	1(102)	0.005
Model 2 (Figure 5b)	LSTM (serial)	Yes	5 layers (51)	None	1(102)	0.005
Model 3 (Figure 6)	Bi-LSTM (serial)	No	5 layers (51)	None	1(102)	0.005
Model 4 (Figure 7a)	MTL, 5 Bi-LSTMs	Yes	1 layer per sub-model (51)	None	2 (1020 & 102)	0.005
Model 5 (Figure 7b)	MTL, 5 gated Bi-LSTMs	Yes	1 layer per sub-model (51)	None	2 (1020 & 102)	0.005
Model 6 (Figure 7c)	MTL, 5 Bi-LSTMs + Conv Layer	Yes	1 layer per sub-model (51)	Kernel (5,2), stride (3,2)	1(102)	0.005
Model 7 (Figure 7d)	MTL, 5 gated Bi-LSTM + Conv Layer	Yes	1 layer per sub-model (51)	Kernel (5,2), stride (3,2)	1(102)	0.001

**Table 4 sensors-22-08452-t004:** Performance of different models in prediction of post-treatment gait trajectories with respect to different diseases.

Model No. and Fig. Reference	Model Type	Spinal Cord Injury (SCI)	Multiple Sclerosis (MS)	Stroke	Cerebral Palsy (CP)	Traumatic Brain Injury (TBI)
		**No. of Patients**				
		**11**	**12**	**9**	**3**	**3**
		**No. of Cycles**				
		**474**	**530**	**322**	**148**	**148**
		**RMSE Mean ± Standard Error**				
		**$R^{2}$ Score**				
Model 1 (Figure 5a)	LSTM (serial)	6.82 ± 0.09	6.89 ± 0.10	8.11 ± 0.19	7.66 ± 0.14	5.87 ± 0.11
		0.65	0.69	0.58	0.71	0.67
Model 2 (Figure 5b)	LSTM (serial)	6.71 ± 0.08	6.77 ± 0.08	8.03 ± 0.19	7.23 ± 0.11	7.63 ± 0.32
		0.61	0.65	0.61	0.70	0.67
Model 3 (Figure 6)	Bi-LSTM (serial)	6.9 ± 0.10	6.38 ± 0.10	7.06 ± 0.18	7.2 ± 0.10	7.78 ± 0.22
		0.72	0.78	0.71	0.78	0.59
Model 4 (Figure 7a)	MTL, 5 Bi-LSTMs	6.26 ± 0.08	**5.8 ± 0.11**	6.99 ± 0.019	6.57 ± 0.12	**5.24 ± 0.13**
		**0.78**	**0.79**	0.72	0.76	**0.73**
Model 5 (Figure 7b)	MTL, 5 gated Bi-LSTMs	6.67 ± 0.08	6.11 ± 0.09	7.73 ± 0.29	6.22 ± 0.14	6.07 ± 0.21
		0.75	0.71	**0.74**	0.78	0.63
Model 6 (Figure 7c)	MTL, 5 Bi-LSTMs + Conv Layer	**5.75 ± 0.08**	6.08 ± 0.12	7.16 ± 0.24	6.2 ± 0.12	6.58 ± 0.14
		0.73	0.74	0.71	0.79	0.64
Model 7 (Figure 7d)	MTL, 5 gated Bi-LSTMs + Conv Layer	6.31 ± 0.12	7.59 ± 0.13	**6.24 ± 0.14**	**6.00 ± 0.14**	7.02 ± 0.07
		0.66	0.70	0.66	**0.80**	0.46

Bold entries denote the lowest average RMSE and maximum $R^{2}$ over all limbs having a given disease.

**Table 5 sensors-22-08452-t005:** Performance of different models in the prediction of post-treatment knee gait with respect to different diseases.

Model No. and Fig. Reference	Model Type	SCI	MS	Stroke	CP	TBI
		**No. of Patients**				
		**11**	**12**	**9**	**3**	**3**
		**No. of Cycles**				
		**474**	**530**	**322**	**148**	**148**
		**RMSE Mean ± Standard Error**				
		**$R^{2}$ Score**				
Model 1	LSTM (serial)	7.73 ± 0.09	8.05 ± 0.11	8.62 ± 0.21	10.16 ± 0.13	6.66 ± 0.09
		0.67	0.65	−0.08	0.70	0.59
Model 2	LSTM (serial)	7.58 ± 0.08	8.26 ± 0.09	7.85 ± 0.17	8.56 ± 0.12	8.05 ± 0.27
		0.72	0.62	0.04	0.70	0.55
Model 3	Bi-LSTM (serial)	8.11 ± 0.13	7.41 ± 0.12	7.77 ± 0.21	7.42 ± 0.11	7.89 ± 0.28
		0.67	0.73	0.16	0.78	0.41
Model 4	MTL, 5 Bi-LSTMs	7.51 ± 0.08	7.23 ± 0.14	7.14 ± 0.018	**6.75 ± 0.11**	5.81 ± 0.13
		**0.76**	**0.77**	0.26	**0.80**	0.61
Model 5	MTL, 5 gated Bi-LSTMs	7.62 ± 0.10	7.23 ± 0.11	8.02 ± 0.25	7.00 ± 13	**5.60 ± 0.06**
		0.71	0.64	0.45	0.77	**0.72**
Model 6	MTL, 5 Bi-LSTMs + Conv Layer	**6.94 ± 0.09**	**6.78 ± 0.14**	7.19 ± 0.25	8.63 ± 0.18	8.24 ± 0.17
		0.69	0.70	**0.48**	0.79	0.62
Model 7	MTL, 5 gated Bi-LSTMs + Conv Layer	8.14 ± 0.12	8.52 ± 0.13	**6.21 ± 0.14**	7.82 ± 0.14	5.94 ± 0.07
		0.59	0.66	0.08	0.78	0.34

**Table 6 sensors-22-08452-t006:** Performance of different models in prediction of post-treatment ankle gait with respect to different diseases.

Model No. and Fig. Reference	Model Type	SCI	MS	Stroke	CP	TBI
		**No. of Patients**				
		**11**	**12**	**9**	**3**	**3**
		**No. of Cycles**				
		**474**	**530**	**322**	**148**	**148**
		**RMSE Mean ± Standard Error**				
		**$R^{2}$ Score**				
Model 1	LSTM (serial)	5.91 ± 0.08	5.73 ± 0.08	7.61 ± 0.16	5.16 ± 0.14	5.09 ± 0.13
		0.52	0.50	**−3.75**	0.49	0.08
Model 2	LSTM (serial)	5.85 ± 0.07	5.29 ± 0.07	8.21 ± 0.21	5.89 ± 0.10	7.22 ± 0.36
		0.19	0.37	−4.48	0.37	0.15
Model 3	Bi-LSTM (serial)	5.69 ± 0.007	5.35 ± 0.07	6.34 ± 0.15	6.99 ± 0.09	5.66 ± 0.15
		0.40	0.68	−3.94	0.44	0.34
Model 4	MTL, 5 Bi-LSTMs	5.01 ± 0.08	**4.38 ± 0.08**	6.85 ± 0.19	6.4 ± 0.12	**4.68 ± 0.13**
		**0.54**	**0.69**	−4.13	0.37	**0.54**
Model 5	MTL, 5 gated Bi-LSTMs	4.56 ± 0.06	5.39 ± 0.10	7.14 ± 0.22	**3.77 ± 0.05**	4.93 ± 0.10
		0.44	0.47	−4.20	**0.50**	0.37
Model 6	MTL, 5 Bi-LSTMs + Conv Layer	5.72 ± 0.06	5.00 ± 0.06	7.44 ± 0.32	5.45 ± 0.14	6.54 ± 0.36
		0.46	0.63	−5.14	0.47	−0.03
Model 7	MTL, 5 gated Bi-LSTMs + Conv Layer	**4.49 ± 0.06**	6.66 ± 0.19	**6.26 ± 0.26**	4.17 ± 0.09	10.63 ± 0.26
		0.25	0.46	−4.85	0.23	−0.83

**Table 7 sensors-22-08452-t007:** Performance comparison of the prediction methods. LinReg and MLinReg correspond to linear regression and multiple LinReg, respectively, in [4,16]. PCA stands for principal component analysis. NN99 and NN01 correspond to feedforward neural networks, respectively, in [33,34].

Model	Knee Flexion	Ankle DorsiFlexion
	$R^{2}$ score	
**Model 5 Stroke** ^†^	0.62	−4.69
**Model 6 Stroke** ^†^	0.49	−4.24
**LinReg Stroke**^†^ [4]	0.24	0.43
	Mean RMSE (${}^{\circ }$)	
**Model 4 CP**	6.8	6.4
**Model 5 CP**	7.0	3.8
**PCA + MLinReg CP** [16]	9.0	7.5
**NN99 CP** [33]	9.7	6.7
**NN01 CP** [34]	9.2	not reported

^†^ Only peak flexion and not over the whole time series.

## Data Availability

Due to the nature of this research, participants of this study did not agree for their data to be shared publicly, so supporting data are not available.

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
