# Peer review of "Treatment Outcome Prediction Using Multi-Task Learning: Application to Botulinum Toxin in Gait Rehabilitation"

_sensors, 2022, doi:10.3390/s22218452_

Round 1

Reviewer 1 Report

The Study of Khan and colleagues aims to employ LSTM based architectures to predict gait patterns after BTX-A muscles treatments using data available in the patient pre-treatment. The use of machine learning models to predict post-treatment outcomes certainly represents a good research direction for two main reasons: as first models can be used for a posteriori evaluation of the beneficial effects of the treatments, secondly such models can be employed in decision support systems for clinicians. For this reason Authors are invited to enlarge the introduction also focusing on these aspects. Thus Authors are invited to review the following manuscripts:

(1)  "A Random Tree Forest decision support system to personalize upper extremity robot-assisted rehabilitation in stroke: a pilot study." 2022 International Conference on Rehabilitation Robotics (ICORR). IEEE, 2022.

(2) "Identification of Neurodegenerative Diseases From Gait Rhythm Through Time Domain and Time-Dependent Spectral Descriptors." IEEE Journal of Biomedical and Health Informatics (2022).

Although these points suggests to invest in this direction, it is important to maintain the investigation as much as possible connected with the real practice. For this reason few points have been highlighted in order to give to the Authors the possibility to improve the quality of the manuscript:

1- Many typos are present in the text, so please take care of this in the revision.

2- Authors reported the RMSE showing low errors. However, this metric can mask overfitting problems of the models. Authors are suggested to report in addition the R squared metric in order to support their discussions. 

3- The idea  of predicting treatment outcome can presents problem due to the impossibility to enclose in the model other aspects in relation to the subject component, i.e. psychological factors, age, stress. Thus outliers (either good or bad in the sense of post-treatment recovery) can hinder the implementation of such models in decision support system, please discuss this point. 

4- Users recorded data in a instrumented environment fusing information of a typical movement analysis laboratory. However it could be important to give focus on inertial measurement units for the acquisition of the data in non instrumented environments. Regarding the goodness of kinematic quantity estimation through IMU sensors, authors are suggested to review the following manuscripts:

(1) "Magnetometer-free sensor fusion applied to pedestrian tracking: A feasibility study." 2019 IEEE 23rd International Symposium on Consumer Technologies (ISCT)

(2) "Single IMU displacement and orientation estimation of human center of mass: A magnetometer-free approach." IEEE Transactions on Instrumentation and Measurement 69.8 (2019): 5629-5639.

Author Response

We are so much thankful to the reviewer for all these observations and suggestions, they helped us in improving our study.

Following are the answers of your comments or suggestions:

The Study of Khan and colleagues aims to employ LSTM based architectures to predict gait patterns after BTX-A muscles treatments using data available in the patient pre-treatment. The use of machine learning models to predict post-treatment outcomes certainly represents a good research direction for two main reasons: as first models can be used for a posteriori evaluation of the beneficial effects of the treatments, secondly such models can be employed in decision support systems for clinicians. For this reason Authors are invited to enlarge the introduction also focusing on these aspects. Thus Authors are invited to review the following manuscripts:

(1)  "A Random Tree Forest decision support system to personalize upper extremity robot-assisted rehabilitation in stroke: a pilot study." 2022 International Conference on Rehabilitation Robotics (ICORR). IEEE, 2022.

(2) "Identification of Neurodegenerative Diseases From Gait Rhythm Through Time Domain and Time-Dependent Spectral Descriptors." IEEE Journal of Biomedical and Health Informatics (2022).

Although these points suggests to invest in this direction, it is important to maintain the investigation as much as possible connected with the real practice. For this reason few points have been highlighted in order to give to the Authors the possibility to improve the quality of the manuscript:

Answer: We have included one of the two papers you suggested for the introduction. Because other papers are focusing on the classification of neurodegenerative diseases, but our study focuses mainly on treatment outcome prediction, we have added all those studies that are working in that direction. 

1- Many typos are present in the text, so please take care of this in the revision.

Answer: All the authors have been checked for typos and grammar, and many changes have been made. 

2- Authors reported the RMSE showing low errors. However, this metric can mask overfitting problems of the models. Authors are suggested to report in addition the R squared metric in order to support their discussions. 

Answer: 

We only reported the RMSE on the test sets to avoid any bias or overfitting. We computed the R2 metric, as you suggested, to assess the variance explanation. We added R2 values into tables 4, 5, and 6 (also the new table 7). 
We updated the materials and methods section as follows:
The coefficient of determination ($R^2$), as follows 
\begin{equation}
 \label{r2_eq}
 R^2 = 1-\frac{\sum_{i}(y_i - \hat{y}_i)^2}{\sum_{i}(y_i - \bar{y}_i)^2},
\end{equation}
can be interpreted as the proportion of
the variance in the dependent variable that is predictable from the independent variables (worst value $-\infty$, best value$=+1$), on the opposite to MSE  which magnifies the error if the model outputs a  very bad prediction (worst value $+\infty$, best value$=0$).

Tables 4, 5, and 6 were updated with the corresponding R2 scores. These values are also reported in the results section (section 3).

We also added discussions about the R2 scores in the corresponding section 4:

"MTL models also gave highest r2, better explaining the total variance of the target than the serial models"

"RMSE are usually higher for the knee than for the ankle despite having higher coefficients of determination. This suggests that models are able to explain the variance of the knee angle better, but the amplitudes in the knee are higher than in the ankle. Moreover, no proposed model was able to adequately explain the variance of ankle angle for patients with stroke (only negative R2 scores)"

"In this case, the R2 score of the proposed method for stroke is better for peak knee flexion but worse for peak ankle dorsiflexion. Since the compared models were not trained and tested with the same databases, this comparison must be taken with caution"

3- The idea  of predicting treatment outcome can presents problem due to the impossibility to enclose in the model other aspects in relation to the subject component, i.e. psychological factors, age, stress. Thus outliers (either good or bad in the sense of post-treatment recovery) can hinder the implementation of such models in the decision support system, please discuss this point. 

Answer: Indeed, these factors play a major role in the rehabilitation and treatment outcome. We added this in the discussion section as a limitation of our method:  
"A limitation of the model is that it does not consider other aspects of the patient, such as psychological factors, age, stress, and social environment, among others, that play a major role in the rehabilitation and thus in the treatment outcome."

4- Users recorded data in a instrumented environment fusing information of a typical movement analysis laboratory. However it could be important to give focus on inertial measurement units for the acquisition of the data in non instrumented environments. Regarding the goodness of kinematic quantity estimation through IMU sensors, authors are suggested to review the following manuscripts:

(1) "Magnetometer-free sensor fusion applied to pedestrian tracking: A feasibility study." 2019 IEEE 23rd International Symposium on Consumer Technologies (ISCT)

(2) "Single IMU displacement and orientation estimation of human center of mass: A magnetometer-free approach." IEEE Transactions on Instrumentation and Measurement 69.8 (2019): 5629-5639.

Answer: 

We have added information on different ways of acquiring joint kinematics at the beginning of the materials and methods section:

"Joint kinematics are typically acquired by optoelectronic systems [8] or inertial measurement unit (IMU) systems [25]."

Also in the caption of figure 2:

"Figure 2. Clinical Gait Analysis. Different types of sensors are used to conduct kinematic and kinetic analyses of locomotion in gait labs. These may include optoelectronic motion capture, force platforms, electromyography, and IMU sensors, among others."

Reviewer 2 Report

- Paper focus on an interesting topic of study adopting machine learning framework. Organization of the paper is good. Theoretical background, basics, equations are well explained. The novelty of the paper is sound which is highlighted with good number of references. References are up to date and properly cited as well. However, 

- The comparison to related works is missing. 

- However, the following statements are necessary but missing;

Institutional Review Board Statement, Informed Consent Statement, Data Availability Statement.

- Author contributions should be made in CRediT Taxonomy and journal format. See https://credit.niso.org/

- I recommend the authors publish their data separately in another article and make publicly available if possible.

Author Response

We are thankful to the reviewer for valuable suggestions and comments that helped us to improve our study. 

Following are the answers of your comments and suggestions:

  • The comparison to related works is missing.

Answer: It is indeed difficult to compare to other works since it is the first time the knee and ankle flexion curves are predicted for this particular treatment (botulinum toxin). However, we computed the error for the peak knee flexion to compare with previous works, and also we compared it to predictions of other treatments (surgery). We added a paragraph in the discussion (section 4) and tables 7 and 8 with the full results of the treatment effect prediction by various approaches.

  • However, the following statements are necessary but missing;

Institutional Review Board Statement, Informed Consent Statement, Data Availability Statement.

Answer: We added this information. There is no informed consent statement because this work is retrospective. Patients were informed and did not oppose data usage.

  • Author contributions should be made in CRediT Taxonomy and journal format. See https://credit.niso.org/

Answer: We have changed it as per your suggestion.

  • I recommend the authors publish their data separately in another article and make publicly available if possible.

Answer: Unfortunately, the internal ethics committee does not authorize patient data sharing.